# The Advantages of Clinical Nutrition Use in Oncologic Patients in Italy: Real World Insights

**DOI:** 10.3390/healthcare8020125

**Published:** 2020-05-06

**Authors:** Paolo Pedrazzoli, Riccardo Caccialanza, Paolo Cotogni, Luca Degli Esposti, Valentina Perrone, Diego Sangiorgi, Francesco Di Costanzo, Cecilia Gavazzi, Armando Santoro, Carmine Pinto

**Affiliations:** 1Medical Oncology Fondazione IRCCS Policlinico San Matteo and Department of Internal Medicine and Medical Therapy, University of Pavia, 27100 Pavia, Italy; p.pedrazzoli@smatteo.pv.it; 2Clinical Nutrition and Dietetics Unit, Fondazione IRCCS Policlinico San Matteo, 27100 Pavia, Italy; 3Pain Management and Palliative Care, Department of Anesthesia, Intensive Care and Emergency, Molinette Hospital, University of Turin, 10126 Turin, Italy; paolo.cotogni@unito.it; 4Clicon S.r.l., Health Economics and Outcomes Research, 48121 Ravenna, Italy; luca.degliesposti@clicon.it (L.D.E.); valentina.perrone@clicon.it (V.P.); diego.sangiorgi@clicon.it (D.S.); 5Medical Oncology Unit, AOU Careggi, 50134 Firenze, Italy; dicostanzofrancesco@icloud.com; 6Clinical Nutrition Unit Fondazione IRCCS Istituto Tumori Milano, 20133 Milan, Italy; cecilia.gavazzi@istitutotumori.mi.it; 7Humanitas Clinical and Research Center IRCCS, Rozzano and Humanitas University, 20089 Milan, Italy; armando.santoro@cancercenter.humanitas.it; 8Medical Oncology Unit, Clinical Cancer Centre IRCCS—AUSL Reggio Emilia, 42122 Reggio Emilia, Italy; carmine.pinto@ausl.re.it

**Keywords:** clinical nutrition, malnutrition, metastasis, oncology, real-world

## Abstract

This retrospective observational study aimed to provide insights on the use of clinical nutrition (CN) (enteral and parenteral feeding) and outcomes in an Italian real-world setting. The data source comes from administrative databases of 10 Italian Local Health Units. Patients diagnosed with malignant neoplasms from 1 January 2010 to 31 December 2015 were included. Metastasis presence was ascertained by discharge diagnosis in the hospitalization database. CN was identified by specific codes from pharmaceutical and hospitalization databases. Two cohorts were created—one for metastatic patients (*N* = 53,042), and one for non-metastatic patients (*N* = 4379) receiving CN. Two survival analyses were set for the cohort of metastatic patients—one included patients receiving CN and the second included malnourished patients. Our findings show that (1) administration of CN is associated with positive survival outcomes in metastatic patients with gastrointestinal, respiratory, and genitourinary cancer; (2) CN in malnourished metastatic patients with gastrointestinal and genitourinary cancer was associated with significant improvement in survival; (3) early administration of CN was associated with improvement in survival in non-metastatic patients with gastrointestinal cancer (HR 95%CI: 0.5 (0.4–0.6), *p*-value < 0.05). This study highlights the need to improve the assessment of nutritional status in oncologic patients and suggests a potential survival benefit of CN treatment in metastatic disease.

## 1. Introduction

Weight loss and malnutrition (underfeeding) have been shown to be negative prognostic factors in cancer patients [1]. In particular, malnutrition, which is common in oncologic patients, may negatively affect patients’ survival, functional status, quality of life, and tolerance to cancer treatments [2,3,4,5,6]. The prevalence of malnutrition varies depending on tumor stage and site [2], and estimates were found to range from 15% to 40% at the time of diagnosis, increasing up to 80% with cancer progression [7]. In addition, research suggests that approximately 20–30% of cancer patients may die as a consequence of malnutrition rather than cancer [8]. International guidelines promote the assessment and management of malnutrition in cancer patients [1], and it is recommended to regularly screen all cancer patients for signs of active or imminent malnutrition [9]. The Global Leadership Initiative on Malnutrition (GLIM) provided global consensus criteria for diagnosing malnutrition in adults in clinical settings—after the identification of “at-risk” status, the diagnosis of malnutrition is assessed after the evaluation of phenotypic (non-volitional weight loss, low body mass index, reduced muscle mass) and etiologic (reduced food intake or assimilation, disease burden/inflammation) criteria [10]. Recently, the American Society for Parenteral and Enteral Nutrition (ASPEN) has recommended artificial nutrition at the earliest opportunity in malnourished patients [6,11]. The Italian guidelines for nutrition management in cancer patients recommend nutritional screening to be performed at defined time points, starting from diagnosis to early identify signs of malnutrition [5,12]. Currently, available clinical nutrition (CN) options for oncologic patients at nutritional risk or with malnutrition include dietary counseling, administering oral nutritional supplements, enteral nutrition (EN), and parenteral nutrition (PN) [9]. The aims of CN interventions are to maintain or improve food intake and mitigate metabolic derangements, to maintain skeletal muscle mass and physical performance, to decrease the risk of reductions or interruptions of scheduled anticancer treatments, and to improve quality of life [4].

Although recommendations on the optimal management of nutritional support for patients with malignancies have been provided [4], malnutrition often stays undiagnosed [13,14], and an important share of malnourished patients does not receive adequate nutritional support [2]. There is a gap between guideline recommendations and clinical practice that can be attributed to the insufficient awareness of nutritional problems among health care professionals [15,16], the lack of structured collaboration between oncologists and CN specialists, and the lack of clinical trials aimed at estimating the optimal nutritional support required in different care settings for cancer patients [5].

Currently, there is limited published evidence about the extent of malnutrition assessment, and of CN use in cancer patients in real-world settings. Furthermore, few studies are published on the impact of CN interventions on cancer patient’s survival. The results from a European multi-country real-world study aimed at investigating malnutrition diagnosis, healthcare resource use, and CN use in cancer patients in France, Germany, and Italy using administrative databases have been recently published [15]. According to this study, in Italy, about 8% of metastatic cancer patients received CN, and among metastatic cancer patients who received CN, only 11% were diagnosed with malnutrition [15]. 

The present manuscript reports the results of an in-depth analysis using the Italian data from the above-mentioned study [15]. In particular, the main focus of this study is to provide further real-world insights on the use of CN (either EN or PN) in oncologic patients and outcomes. 

## 2. Materials and Methods 

### 2.1. Data Source

A retrospective observational study was performed based on the administrative databases from 10 Italian Local Health Units (LHUs), geographically distributed throughout the national territory. For this study, the following databases were used: (i) beneficiaries database including patients characteristics; (ii) outpatient pharmaceuticals database which includes Anatomical Therapeutic Chemical (ATC) codes of drugs dispensed, number of and costs per packages, prescription date, and drug package ID; (iii) hospitalization database which provided all the hospitalization data, including primary and secondary diagnoses recorded using the International Classification of Diseases, Ninth Revision, Clinical Modification (ICD-9-CM); (iv) laboratory test and specialist visit database, including the date and type of laboratory tests or specialist visit; (v) exemption database.

Administrative databases have been shown to be a powerful tool in supporting conventional methods used in epidemiological studies [17,18,19,20,21]. Each patient was identified by an anonymous code, which permitted the electronic linkage between these databases. Data were anonymized in full compliance with the European General Data Protection Regulation (GDPR) (2016/679). No identifiers related to patients were provided to the researchers. All the results of the analyses were produced as aggregated summaries, which are not possible to assign, either directly or indirectly, to individual patients. Informed consent was not required for using encrypted retrospective information. According to Italian law, this study has received approvals from the local ethics committees of the LHUs involved in the study [22]. The name of the ethics committees that approved the study, along with the number/ID of the approval, are reported in the Appendix A.

### 2.2. Study Population

All patients with a diagnosis of malignant neoplasm during the enrollment period (1 January 2010 to 31 December 2015) were included. Malignant neoplasm diagnosis was ascertained by discharge diagnosis for malignant neoplasm (ICD-9-CM codes: 140–209, 235–239) in the hospitalization database. Among the included patients, the presence of metastasis was confirmed by the diagnosis of a secondary malignant neoplasm (ICD-9-CM codes: 196–199) in the hospitalization database during the enrollment period. CN was defined as the presence of multivitamin agents (ATC code: A11BA), solutions for PN (ATC code: B05BA), intravenous additive solution (ATC code B05XB, B05XC) for nutrition at home, and enteral infusion (ICD-9-CM code: 96.6) or parenteral infusion (ICD-9-CM code: 99.15) of concentrated nutritional substances for nutrition in the hospital. 

The date of inclusion in the study, defined as the index date (ID), corresponded to the date of first CN evidence for non-metastatic patients and to the date of first metastasis diagnosis for metastatic patients during the enrollment period. Patients were characterized during the twelve-month period preceding the ID (the characterization period) for the collection of demographic information and data about comorbidities and diagnosis of malnutrition (ICD-9-CM: 260–269). Patients were further selected when presenting a diagnosis of the following cancer during the characterization period—Malignant Head and Neck (ICD-9-CM codes 140–149), gastrointestinal (ICD-9-CM codes 150–159), respiratory (ICD-9-CM codes 160–165), genitourinary (ICD-9-CM codes 179–189), and hematology (ICD-9-CM codes 200–208). Patients were followed-up for twelve months, starting on the ID (the follow-up period). Information collected during follow-up included evidence of CN, death, chemotherapy (identified by the ATC code: L01 or ICD-9-CM code: V58.1), drug prescriptions, and hospitalizations for all the included patients.

Two cohorts were created—one for metastatic patients, and one for non-metastatic patients receiving CN. In each cohort, patients were grouped depending on the cancer site, and all the analyses were separately run on each group, unless differently specified. 

#### 2.2.1. Metastatic Patients

The cohort of metastatic patients initially included all the patients who had at least one diagnosis of metastasis during the enrollment period; the date of the first metastasis diagnosis during the enrollment period was considered as the ID. Patients were excluded from the analysis when at least one of the following conditions occurred: the presence of at least one prescription of solution for PN containing glucose at low concentrations (<20%) during the characterization period and during follow-up, the presence of at least one metastasis diagnosis during the characterization period, the evidence of CN during the characterization period. 

#### 2.2.2. Non-Metastatic Patients Receiving CN

Patients without metastasis diagnosis during the entire study period were defined as non-metastatic patients. The cohort of non-metastatic patients initially included all the patients who had evidence of CN during the enrollment period; the date of the first CN evidence during the enrollment period was considered the ID. 

### 2.3. Outcomes Definition and Statistical Analysis 

A first cross-tabulation of patients’ demographics and clinical characteristics was presented for the total cohort of metastatic patients, with stratification by presence/absence of CN evidence. The characteristics included variables such as age, gender, and a modified version of the Charlson Comorbidity Index [23] (mCCI). CCI assigns a score to each concomitant disease identified through treatments and hospitalizations during the characterization period; in this study, a modified version not accounting for cancer was used. Then, focusing on metastatic patients who had evidence of CN, a second cross-tabulation of patients’ characteristics (age, gender, and mCCI) stratified by the time of administration of CN was presented. In order to define time of administration of CN, for each cancer subgroup, the time (months) since metastasis diagnosis to first CN prescription was calculated for all metastatic patients receiving CN. Patients were then classified as having an early or late administration, defined below.

Two different survival analyses were set for the cohort of metastatic patients during the follow-up period. The first one involved the subgroup of patients receiving CN who had at least three months of follow-up and evaluated the effect of early versus late administration of CN on the survival since metastasis diagnosis. The second survival analysis involved the subgroup of metastatic patients diagnosed with malnutrition which had at least three months of follow-up and evaluated the effect of the administration of CN versus no administration on survival since metastasis diagnosis. Survival analyses did not account for patients with hematology cancer as numbers were too low. As for the cohort of non-metastatic patients, survival analyses were set as months from cancer diagnosis to death. The effect on survival since cancer diagnosis coming from an early administration of CN versus a late administration was evaluated.

In order to define early and late clinical administration, time (months) since cancer diagnosis to first CN prescription was calculated on all non-metastatic patients receiving CN for each diagnosis group; patients were then classified as having an early or late administration when the CN administration occurred below or above the first quartile, respectively. 

Kaplan–Meier curves plotting the time to death since cancer diagnosis or since metastasis were presented for head and neck, gastrointestinal, respiratory, genitourinary, and hematology cancer patients (for the non-metastatic cohort only). Also, multivariate Cox regression models were run to provide hazard ratios for death, with the time of CN administration (early versus late) being the independent variable, while age, gender, mCCI, and chemotherapy presence/absence being model covariates.

All statistical analyses were performed using STATA SE, and a *p*-value < 0.05 was considered statistically significant. 

## 3. Results

Patients who met the criteria for inclusion in the cohort of metastatic patients were 53,042. The proportion of metastatic patients who received CN was small for all the considered cancer types and ranged from 5% of hematology cancer patients to 13% of head and neck cancer patients. No noticeable differences depending on CN administration were observed in metastatic patients, whose characteristics are reported in Table 1. The mean age (SD) of metastatic patients was 67.2 (12.4) for the ones receiving CN, 70.1 (12.2) for those without CN. Overall, men accounted for higher proportions, no matter having received CN or not, except for the genitourinary subgroup, for which females accounted for a higher proportion than males among patients receiving CN (Table 1). Among the metastatic patients who received CN, those who also underwent chemotherapy represented about 5%. In particular, patients with evidence of chemotherapy during follow-up, focusing on those who had evidence of CN, accounted for 3.3% of respiratory cancer patients, for 3.8% of genitourinary cancer patients, for 4.9% of gastrointestinal cancer patients, for 8.9% of head and neck cancer patients, and for 14.4% of hematology cancer patients. The characteristics of metastatic patients who received CN and who had at least three months of follow-up stratified by early and late administration of CN are reported in Appendix A. 

Kaplan–Meier curves plotting the time to death since metastasis diagnosis in the two groups of early and late CN patients are reported in Figure 1, while the results from the multivariate Cox regression models that confirmed those from Log-Rank tests are synthetized in Appendix A.

Overall, the proportion of patients who had a malnutrition diagnosis during the characterization period were low, even if it was higher among those patients who received CN, as shown in Appendix A—among the cancer subgroups, head and neck was the one with the highest proportion of patients receiving malnutrition diagnosis. The differential diagnoses of malnutrition in the cohort of metastatic patients are provided in Appendix A.

Figure 2 shows Kaplan–Meier curves comparing survival time since metastasis diagnosis for malnourished metastatic patients with CN versus without CN. Appendix A reports the results from the Cox multivariate regression models comparing death risk in malnourished metastatic patients with CN versus without CN. 

The cohort of non-metastatic patients who received CN was composed of 4379 patients, and their characteristics are reported in Appendix A. Similar to the cohort of metastatic patients, men accounted for higher proportions than women, particularly for the respiratory and genitourinary cancer subgroups, that were also the most health-impaired ones. The proportion of patients with a diagnosis of malnutrition was still small, even if they were higher than those observed among metastatic patients. 

Focusing on non-metastatic patients who had at least three months of follow-up, we observed a higher proportion of subjects who had an early administration of CN than those observed for metastatic patients (Appendix A). Kaplan–Meier curves plotting the time to death since cancer diagnosis in the two groups of early and late CN patients are reported in Figure 3. The results from multivariate Cox regression models confirmed that non-metastatic gastrointestinal cancer patients who received early CN were less likely to die during follow-up when compared to those who had late CN (Appendix A).

## 4. Discussion

The main finding from this study is that the assessment of malnutrition and the administration of CN in oncologic patients is not common practice in Italy, mainly due to the lack of standardized indications for different cancer types. In fact, the data analysis from this study suggests that oncologic patients, particularly those with gastrointestinal cancer, might benefit from the administration of CN. In particular, the findings suggest that (1) administration of CN is associated with positive survival outcomes in metastatic patients with gastrointestinal, respiratory, and genitourinary cancer, while no benefit was observed for concomitant chemotherapy (2) CN in malnourished metastatic patients with gastrointestinal and genitourinary cancer was associated with significant improvement in survival, and this effect was comparable to that of concomitant chemotherapy, (3) early administration of CN was also associated with improvement in survival in non-metastatic patients with gastrointestinal cancer, while no benefit was observed for concomitant chemotherapy.

Malnutrition is a highly prevalent condition among oncologic patients [2,3,4,5,6,7,9], particularly for those with advanced cancer [9]. The currently available literature suggests a prevalence of malnutrition that ranges from 30 to 60% in gastrointestinal cancer patients [2,24,25,26], and can be significantly higher in specific cancers (e.g., 85% in pancreatic cancers) [27]. Hence, the small proportion of patients reported with malnutrition diagnosis is not likely to reflect an actual low prevalence of the condition, rather an infrequent practice of oncologic patients’ nutritional assessment. Our findings confirm similar results from a recent survey conducted in Italy, which found that nutritional counseling was available only for 15% of hospitalized cancer patients, and only 26% received nutritional information by the oncologists [28], and from a Belgian study, which showed a very low rate of nutritional support in cachectic patients [29].

The insufficient awareness towards nutritional management on cancer patients is corroborated by the low rate of patients receiving CN, which is in line with a recent multicenter French study [2]. It is also worth mentioning that most metastatic patients with CN administration did not receive early treatment. Considering this evidence and the available literature, we may speculate that the underperformed assessment of malnutrition and the underutilization of CN in oncology is not an exclusive alarming Italian story—it represents a widespread emergency that should be regarded at the European level [30].

The improvement of oncologic patients’ treatment with regard to their nutritional status should be promoted, as the close link between malnutrition and negative prognostic factors on several outcomes in cancer has already been demonstrated [1,2,3,4,5,6]. A recent study by Caccialanza et al. evaluated the effects of an early weekly administration of supplemental PN (SPN) in hypophagic hospitalized cancer patients at nutritional risk on a set of parameters, including body composition. In the absence of any relevant clinical complication, a strictly monitored early short-term SPN resulted in the improvement of the parameters [6]. A study by Chen and colleagues retrospectively compared the clinical and economic benefits derived from the implementation of a nutritional support program with routine care in unresectable locally advanced esophageal squamous cell carcinoma patients who underwent concomitant chemoradiotherapy. The nutritional support group had significant advantages over the routine care with respect to overall response rate, biochemical parameters of albumin, and lengths of hospital stay [31]. A study by Trestini and colleagues [1] evaluated the prognostic value tailored nutritional counseling in patients affected by pancreatic ductal adenocarcinoma during chemotherapy, which resulted in being a significant predictor for better overall survival.

Our study has some limitations to be acknowledged, which principally consist of the retrospective and descriptive nature of the analysis, which was based on data collected through administrative claims databases. First, such databases do not provide clinical information useful for the definition of malnutrition according to the Global Leadership Initiative on Malnutrition (GLIM) criteria; therefore, malnourished patients could be underestimated as only patients with discharge diagnosis for malnutrition were identified. However, the identification of malnutrition by ICD-9-CM codes is reliably used in real-world studies [32,33]. Another limitation is the lack of data on the specific type of nutritional support provided, as well as on relevant clinical outcome measures, such as the effectiveness of treatment and disease severity, comorbidities, and other potential confounders that could have influenced the results. Diagnosis of malnutrition in metastatic patients has been searched during the characterization period, but not during follow-up; therefore, the proportions of metastatic patients with a diagnosis of malnutrition may be underestimated. The analysis evaluating the effect of early versus late administration of CN on survival since cancer diagnosis for non-metastatic patients did not necessarily consider the first diagnosis of cancer for each patient as a starting point for the patient’s observation. On the other hand, the main strength of the present study is the validity of the overarching source of information used. Administrative databases represent a valuable source for recording and monitoring consumptions that are at the expense of the Italian National Health Service (NHS), as they give information on most services provided in a healthcare environment. Furthermore, conducting studies based on secondary-data is a suitable approach for assessing important outcomes in the real-world setting [34].

These findings highlight the urgent need for an improvement in the assessment of nutritional status in oncologic patients, which is in line with calls for routine screening and recommendations from the international guidelines. Our analysis provides an indication for apparent benefits of CN in general and earlier CN treatment, particularly in patients with gastrointestinal cancer, who are at high risk of developing malnutrition. The combined evaluation of our results with the available literature suggests that awareness programs toward this issue should be implemented at the European level.

## Figures and Tables

**Figure 1 healthcare-08-00125-f001:**
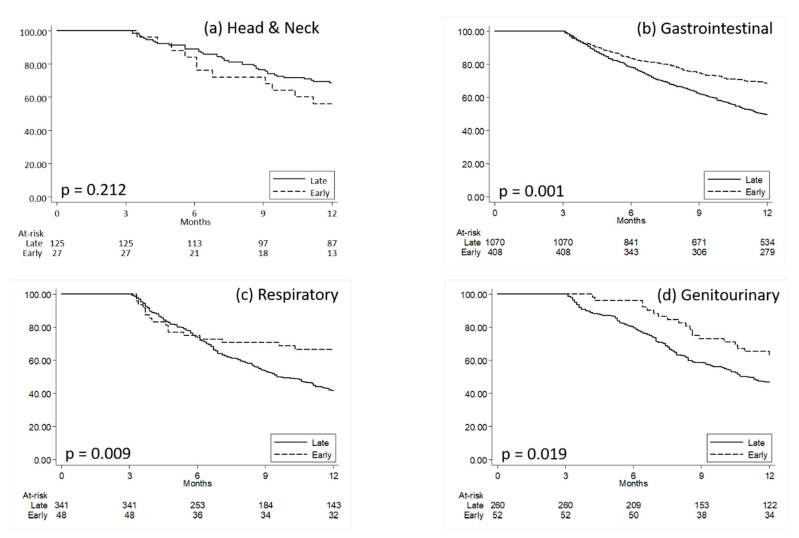
Kaplan–Meier curves comparing survival time since metastasis diagnosis for metastatic patients with early versus late clinical nutrition. (**a**) Head and Neck; (**b**) Gastrointestinal, (**c**) Respiratory, (**d**) Genitourinary. Quartiles of time (months) since metastasis diagnosis to first clinical nutrition prescription was calculated on all metastatic patients receiving clinical nutrition; patients were then classified as having an early or late administration when presenting a time since metastasis diagnosis to clinical nutrition administration below or above the first quartile, respectively. P: *p*-values from Log-Rank Tests.

**Figure 2 healthcare-08-00125-f002:**
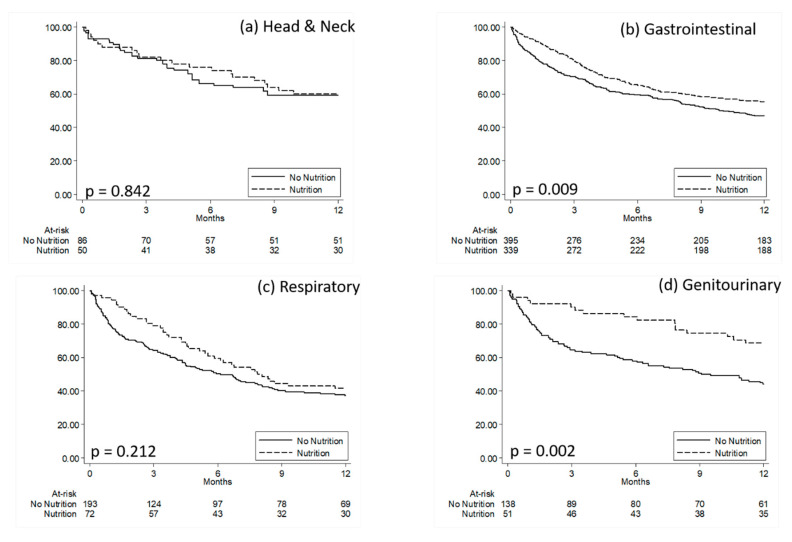
Kaplan–Meier curves comparing survival time since metastasis diagnosis for malnourished metastatic patients with clinical nutrition versus malnourished metastatic patients without clinical nutrition. (**a**) Head and Neck; (**b**) Gastrointestinal, (**c**) Respiratory, (**d**) Genitourinary. P: *p*-values from Log-Rank Tests.

**Figure 3 healthcare-08-00125-f003:**
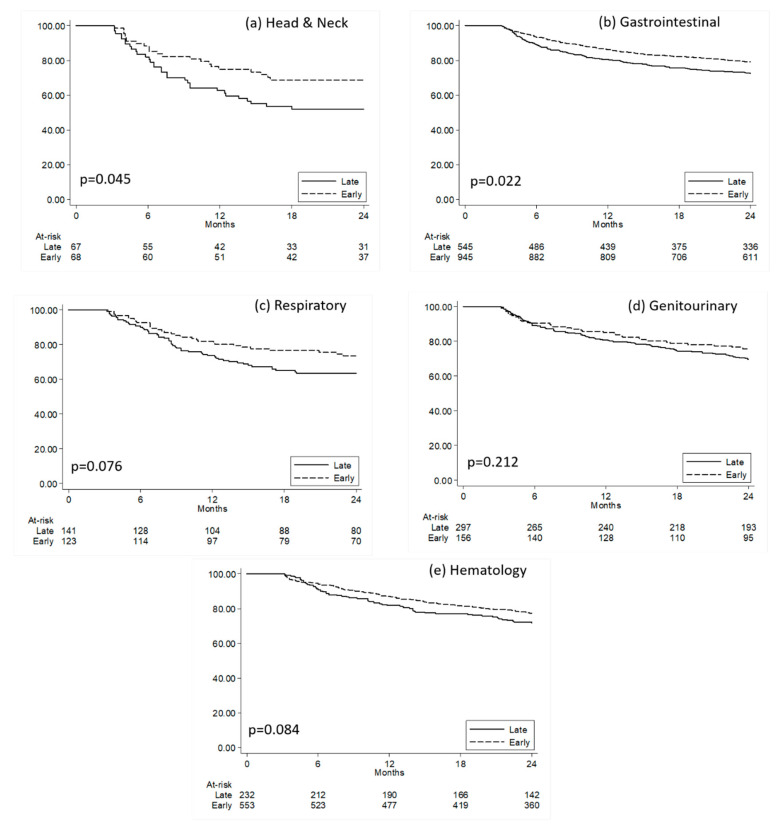
Kaplan–Meier curves comparing survival time since cancer diagnosis for non-metastatic patients with early versus late clinical nutrition. (**a**) Head and Neck; (**b**) Gastrointestinal, (**c**) Respiratory, (**d**) Genitourinary, (**e**) Hematology. Quartiles of time (months) since cancer diagnosis to the first clinical nutrition prescription was calculated in all non-metastatic patients receiving clinical nutrition; patients were then classified as having early or late administration when presenting a time since diagnosis to clinical nutrition administration below or above the first quartile, respectively. P: *p*-values from Log-Rank Tests.

**Table 1 healthcare-08-00125-t001:** Characteristics of metastatic patients according to cancer site and stratified by presence/absence of clinical nutrition.

	Metastatic Patients with CN (*N* = 4814)				Metastatic Patients without CN (*N* = 48,228)			
	*N* (%) ^1^	Age Mean (SD)	Men *N* (%) ^2^	mCCI ^3^ Mean (SD)	*N* (%) ^1^	Age Mean (SD)	Men *N* (%) ^4^	mCCI ^3^ Mean (SD)
All Clusters (*N* = 53,042)	4814 (9.1%)	67.2 (12.4)	2925 (60.8%)	1.0 (1.0)	48,228 (90.9%)	70.1 (12.2)	30,259 (62.7%)	1.0 (1.0)
Head and Neck (*N* = 2141)	271 (12.7%)	62.2 (12.0)	199 (73.4%)	0.8 (1.0)	1870 (87.3%)	65.3 (12.7)	1387 (74.2%)	0.9 (0.9)
Gastrointestinal (*N* = 23,667)	2937 (12.4%)	67.6 (12.5)	1776 (60.5%)	1.0 (1.0)	20,730 (87.6%)	70.8 (12.0)	11,685 (57.2%)	0.9 (1.0)
Respiratory (*N* = 14,475)	814 (5.6%)	67.2 (10.8)	615 (75.6%)	1.3 (1.1)	13,661 (94.4%)	69.7 (11.0)	10,063 (73.7%)	1.2 (1.0)
Genitourinary (*N* = 10,983)	702 (6.4%)	67.6 (13.2)	287 (40.9%)	1.0 (1.0)	10,281 (93.6%)	70.4 (13.2)	5997 (58.3%)	1.0 (0.9)
Hematology (*N* = 1776)	90 (5.1%)	67.5 (17.7)	48 (53.3%)	1.2 (1.2)	1686 (94.9%)	68.5 (14.4)	947 (56.2%)	1.0 (1.0)

^1^ (First row) Proportions calculated over the total number of metastatic patients. (Following rows) Proportions calculated over the total number of metastatic patients with the corresponding cancer type. ^2^ (First row) Proportions calculated over the total number of metastatic patients who received clinical nutrition. (Following rows) Proportions calculated over the total number of metastatic patients with the corresponding cancer type who received clinical nutrition. ^3^ mCCI. Modified Charlson Comorbidity Index, not accounting for cancer. ^4^ (First row) Proportions calculated over the total number of metastatic patients who did not receive clinical nutrition. (Following rows) Proportions calculated over the total number of metastatic patients with the corresponding cancer type who did not receive clinical nutrition.

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
