# Peer review of "The Advantages of Clinical Nutrition Use in Oncologic Patients in Italy: Real World Insights"

_healthcare, 2020, doi:10.3390/healthcare8020125_

Round 1

Reviewer 1 Report

Thank you for the opportunity to review this manuscript titled “Extent And Advantages Of Clinical Nutrition Use In Oncologic Patients In Italy: Real World Insights”.

This Italian retrospective study aimed to provide real-world insights on the importance of the use of clinical nutrition to improve survival outcomes of cancer patients. Italian databases of metastatic and non-metastatic patients from several Italian LHUs are used and in-depth analysed. This study showed a significant improvement in survival outcomes by clinical nutrition in metastatic patients (particularly gastrointestinal cancer patients) and in malnourished metastatic patients with gastrointestinal and genitourinary cancer. Moreover, this manuscript highlights that, although recommendations guidelines on malnutrition assessment and nutritional support are provided, malnutrition is still underappreciated and clinical nutrition is still underutilized by oncologists in Italy (and globally in Europe).

This study is very similar in methods and results to a recent publication of some of the Authors (Unmet needs in clinical nutrition in oncology: a multinational analysis of real-world evidence. Caccialanza R, Goldwasser F, Marschal O, Ottery F, Schiefke I, Tilleul P, Zalcman G, Pedrazzoli P. Ther Adv Med Oncol. 2020 Feb 14;12:1758835919899852. doi: 10.1177/1758835919899852. eCollection 2020)

This article is well-written. The results (figures and tables, suppl. files) and the discussion are very interesting and relevant.

Please see only some suggestions below:

Abstract:

L29: diagnosed with malignant neoplasm

L34: Diagnosed with malnutrition

L34: The period ‘.in which outcome of CN vs no-CN administration..’ is not very clear.

Introduction:

L54-55: ‘International guidelines promote the assessment … of malnutrition.’   Recently, the GLIM (convened by several of the major global clinical nutrition societies) built an international consensus on criteria for the diagnosis of malnutrition for adults in clinical settings [Cederholm et al., GLIM criteria for the diagnosis of malnutrition - A consensus report from the global clinical nutrition community. Clin Nutr. 2019 Feb;38(1):1-9. doi: 10.1016/j.clnu.2018.08.002 ]. Please precise the definition of malnutrition.

Methods

L117: the definition of the characterization period is not very clear. Could you explain what the index date (ID) is?

L123: The diagnosis of Malnutrition according to codes ICD-9-CM: 260-269 includes a group of syndromes ranging from Kwashiorkor to Vitamin D deficiency. It seems a very broad definition, somewhat distant from that of the GLIM criteria. What were the differential diagnoses within the cohort? It should be advisable to put it into supplementary files, excluding the deficit of vitamins from the analysis.

Moreover, the poor overlap between the definition of malnutrition (according to GLIM Criteria) and that of ICD-9-CM codes should be acknowledged and discussed as a potential limitation of the study.

L157 -158: the sentence is unclear, could you please reformulate.

L166- 167: the sentence is unclear, could you please reformulate.

Results

L224-225: please reformulate this period. Do you mean ‘comparing survival time since metastasis diagnosis for malnourished metastatic patients with CN versus without CN’ ?

L226-228: Table S3 reports “the results…with that of those not receiving CN” Could you replace this period with the caption of Table S3 in suppl file?

Minor revisions:

Please replace ‘&’ with ‘and’ (head and & neck cancer L159, L175, L193, etc.., tables and figures)

L313 National Health Service (NHS)

Reviewer 2 Report

Good work on the discussion and results part. However, the introduction and methods section are very confusing and need English editing. I find myself reading the same paragraph a couple of times to fully comprehend what you are trying to say. Additionally, you need to backup your choice of methods and explain why you choose every step of your study design. All other comments are in the word doc. 

Round 2

Reviewer 2 Report

This paper has come out nicely. I only had 1 minor comment, please add the standard deviation when you report a mean (see comment in manuscript)

Kind regards 

Author Response

We thank the Reviewer for all his/her suggestions. We have now added the standard deviation when reporting means.